# Serum Interleukin-17 and Its Association with Inflammation and Bone Remodeling in Rheumatoid Arthritis and Hand Osteoarthritis: Insights from Musculoskeletal Ultrasound

**DOI:** 10.3390/diagnostics15111335

**Published:** 2025-05-26

**Authors:** Amany M. Ebaid, Essam Atwa, Mohamed A. Mortada, Hibah Abdulrahim Bahri, Noura Almadani, Noha M. Hammad

**Affiliations:** 1Rheumatology and Rehabilitation Department, Faculty of Medicine, Zagazig University, Zagazig 44519, Egypt; etama958@gmail.com (E.A.); m_a_mortada@yahoo.com (M.A.M.); 2Medical-Surgical Department, College of Nursing, Princess Nourah bint Abdulrahman University, Riyadh 84428, Saudi Arabia; habahri@pnu.edu.sa; 3Community and Psychiatric Mental Health Nursing Department, College of Nursing, Princess Nourah bint Abdulrahman University, P.O. Box 84428, Riyadh 11671, Saudi Arabia; naalmadani@pnu.edu.sa; 4Department of Medical Microbiology and Immunology, Faculty of Medicine, Zagazig University, Zagazig 44519, Egypt; nmhammad@medicine.zu.edu.eg; 5Al Jouf Regional Laboratory, Sakaka 72345, Saudi Arabia

**Keywords:** AUSCAN, HAQ, IL-17, MSUS, osteoarthritis, rheumatoid arthritis

## Abstract

**Objectives**: The objective of this study was to evaluate the relationship between interleukin-17 (IL-17) serum levels, musculoskeletal ultrasound (MSUS) observations, and clinical disease activity in patients with rheumatoid arthritis (RA) and hand osteoarthritis (OA). **Methods**: This case–control study involved 120 participants, with 40 individuals assigned to each of the three groups: RA, OA, and control. IL-17 serum levels were quantified in all participants. MSUS of the hand joints was performed on all RA and OA patients. Disease activity in patients with RA was assessed using the Clinical Disease Activity Score (CDAS). Both RA and OA patients completed a Visual Analog Scale (VAS) to evaluate pain intensity. Functional status was evaluated using the Health Assessment Questionnaire (HAQ) for RA patients, while the Australian/Canadian (AUSCAN) Osteoarthritis Hand Index was utilized for OA patients. **Results**: Serum levels of IL-17 were significantly higher in both the RA and OA groups compared to the control group. Among RA patients, a positive correlation was identified between the CDAS and the VAS for pain. In OA patients, a significant correlation was observed between VAS scores and serum IL-17 levels. Additionally, serum IL-17 levels were associated with the presence of synovitis in both RA and OA groups; however, no significant association was found between IL-17 levels and bony changes such as erosions or osteophytes. In terms of functional evaluation, serum IL-17 levels correlated with HAQ in the RA group, but not with AUSCAN in the OA group. **Conclusions**: Elevated IL-17 serum levels are linked to inflammatory changes identified by MSUS but not to bony changes. These findings suggest that the rise in IL-17 levels in both OA and RA is primarily driven by underlying inflammatory processes, positioning IL-17 as a potential therapeutic target for both conditions.

## 1. Introduction

The interleukin-17 (IL-17) family includes six members (IL-17A-F) that exert their action through five types of receptors (IL-17RA-E) [1,2]. IL-17, secreted by CD4+ T helper (Th) 17 and CD8+ cytotoxic T (Tc) 17 cells, is involved in the pathogenesis of many inflammatory conditions and autoimmune diseases [3,4].

In recent years, the pleiotropic inflammatory cytokine interleukin-17 (IL-17) has been the focus of extensive research due to its involvement in both degenerative and autoimmune joint diseases. It has been particularly highlighted in rheumatoid arthritis (RA), osteoarthritis (OA) [5], ankylosing spondylitis [6], and psoriatic arthritis (PsA) [7]. Elevated circulating levels of IL-17 have been shown to contribute significantly to the onset and progression of inflammatory arthritis, supporting its relevance as a key mediator in disease pathophysiology [8].

In the context of RA, autoreactive Th1 and Th17 cells are known to play a central role in disease pathogenesis [9]. The earliest pathological changes primarily occur in the synovial membrane, which becomes heavily infiltrated by these inflammatory immune cells. This infiltration leads to elevated levels of IL-17 in both the serum and synovial fluid of patients with RA [3].

In OA, although the evidence is more limited compared to RA, Kamel et al. demonstrated that IL-17 levels in the synovial fluid of knee OA patients correlated with radiographic progression, highlighting the potential inflammatory subset of OA that extends beyond the typical mechanical wear and tear [10].

Further, the heterogeneous expression patterns of the IL-17 family and their receptors in synovial tissues show great variability among individual patients with RA, PsA, and OA [11,12]. This variability may explain the inconsistent clinical responses to anti-IL-17 therapy. Therefore, stratifying patients based on IL-17 expression levels may enhance the efficacy of targeted treatment, underscoring the need for biomarker-driven therapeutic approaches [11].

Musculoskeletal ultrasound (MSUS) is now considered a very popular and accurate method for both diagnosis and follow-up in rheumatology clinics [13]. Both grayscale (GS) and power Doppler (PD) ultrasound (US) are sensitive to change and able to predict arthritis development and radiographic structural damage [14]. By virtue of the higher axial and lateral resolution of US, destructive and/or hypertrophic changes on the bone surface and even tiny abnormalities can be detected easily earlier than their appearance on plain X-rays or even magnetic resonance imaging [15]. Recently, the Outcome Measures in Rheumatology (OMERACT) have redefined, updated, and validated the definitions of US pathologies, elementary lesions, and scoring systems in rheumatology (e.g., synovitis, osteophytes, etc.) [16]. MSUS is a promising predictive tool for the development of clinical arthritis, providing the chance to ameliorate risk stratification and disease prevention [17,18].

Although the role of IL-17 in RA and OA is well established, its imaging correlates, particularly those detected by MSUS, remain poorly understood. In this study, we investigated the relationship between serum IL-17 levels and clinical disease scores in patients with RA and OA. Additionally, we examined the correlation between IL-17 levels and MSUS findings in the small joints of the hands in both conditions, with the goal of improving patient stratification based on IL-17-MSUS associations. While a few comparative studies have explored IL-17 expression in RA and OA, particularly in synovial tissue and serum [11], direct comparisons incorporating MSUS findings remain limited. This approach may help identify patient subgroups that would benefit from targeted anti-IL-17 therapy.

## 2. Materials and Methods

This study was conducted at the outpatient clinic of the Rheumatology and Rehabilitation Department of Zagazig University Hospitals and the Immunology Research Laboratory, Department of Medical Microbiology and Immunology, Faculty of Medicine, Zagazig University, Zagazig, Egypt.

### 2.1. Sample Size

Assuming a mean IL-17 level of 204.1 ± 33.8 pg/mL in RA patients [19] and 177.2 ± 41.7 pg/mL in OA patients [20], the minimum required sample size was calculated to be 78 participants. These were equally divided into three groups—RA, OA, and healthy controls—with 26 subjects in each group. The sample size was calculated using Epi software version 6, with a confidence interval (CI) of 95% and a test power of 80%.

### 2.2. Subjects

The present study comprised 120 subjects allocated into three equal groups (RA, OA, and healthy control). RA and OA patients were selected from those attending the Rheumatology and Rehabilitation Outpatient Clinic at Zagazig University Hospitals, Zagazig, Egypt by systematic random sampling during the period from December 2023 to November 2024.

Patients in the RA group met the classification criteria endorsed by the 2010 American College of Rheumatology (ACR). A patient who scores at least 6 points in an established classification system is considered to have RA [21]. Patients in the OA group met the 1990 ACR classification criteria for hand OA. The classification criteria include (1) hand pain, aching, or stiffness; (2) hard tissue enlargement of 2 or more of 10 selected joints and (3) fewer than 3 swollen metacarpophalangeal (MCP) joints; and (4) either hard tissue enlargement in two or more distal interphalangeal (DIP) joints or deformity of 2 or more of 10 selected joints [22].

All control group subjects were healthy volunteers, matched for age and sex with the OA group and for age with the RA group, with no symptoms or signs of RA or OA. Sex matching with the RA group was not feasible due to the higher prevalence of RA in females. Patients suffering from liver impairment, endocrine disorders, coronary heart disease, renal insufficiency, or other inflammatory conditions were excluded from the study.

### 2.3. Clinical History and Health Assessment

A full history was taken from the study participants. In RA patients, the functional status of activities regarding daily living was assessed using the Health Assessment Questionnaire Disability Index (HAQ-DI) [23]. On the other hand, the Australian/Canadian (AUSCAN) Osteoarthritis Hand Index questionnaire was used to assess health status and health outcomes in OA patients [24]. HAQ-DI and AUSCAN were reported by patients under the supervision of trained rheumatologists.

### 2.4. Physical Examination

All the study participants underwent a thorough clinical examination. Pain intensity in RA and OA patients was assessed using the Visual Analog Scale (VAS) [25]. Disease activity in RA patients was measured using the Clinical Disease Activity Score (CDAS), a validated and widely used index that measures overall disease activity on a scale from 0 to 10. This index takes into account both the patient’s and physician’s overall assessment of disease activity (over the last 48 h), the number of swollen and painful joints, and acute-phase reactants, most commonly ESR. The resulting score ranges from 0 to 76 points [21,26]. VAS and CDAS were administered and calculated by trained rheumatologists during clinical visits.

### 2.5. Musculoskeletal Ultrasound Examination

All hands joints (wrists, metacarpophalangeal, proximal interphalangeal, and distal interphalangeal joints) were examined by a HITACHI-ALOKA F3 unit (Guangzhou Rongtao Medical Technology Co., Ltd., Guangzhou, China), a fully digital portable US system using B-mode with linear probes (frequency 10–18 MHz). Patients were positioned in a sitting posture with both hands resting on the examination table or their lap [27]. All patients had a real-time grayscale US. A fixed protocol, following the technical guidelines of the European Society of Musculoskeletal Radiology for wrist scanning, was applied. Joints were examined using MSUS for the inflammatory domain (synovitis, effusion) or the structural damage domain (erosion, osteophytes) following the updated OMERACT for pathological findings in rheumatic diseases [16].

### 2.6. Blood Sampling

Three milliliters of peripheral blood was obtained from each study participant by venipuncture. The samples were centrifuged at 3000 RPM for 10 min to separate the serum, which was subsequently collected and stored at −20 °C for the subsequent quantification of IL-17A serum levels.

### 2.7. Quantitation of IL-17A Serum Level

The sera of all study participants were analyzed for IL-17A levels using a sandwich enzyme-linked immunosorbent assay (ELISA), performed according to the manufacturer’s instructions (Human Interleukin 17 ELISA Kit, Inova, Beijing, China, Catalogue No. In-Hu2141). The assay had a detection range of 2.8 to 200 pg/mL. Samples with concentrations exceeding the assay’s upper limit were diluted appropriately and reanalyzed to obtain accurate measurements. Absorbance was measured at a wavelength of 450 nm using an ELISA reader (Stat Fax^®^ 303 Plus, Awareness Technology, Inc., Palm City, FL, USA). The optical density values were directly proportional to the concentration of IL-17A. Final IL-17A concentrations (in pg/mL) were calculated by comparing the optical density readings to a standard calibration curve.

### 2.8. Statistical Analysis

Data were analyzed using the Statistical Package for the Social Sciences (SPSS), version 22. Continuous variables were presented as mean ± standard deviation (SD), median, and interquartile range (IQR). The Shapiro–Wilk test was employed to assess the normality of continuous variables. One-way analysis of variance (ANOVA) was used to compare continuous variables across independent groups, and the independent t-test or Mann–Whitney U test was applied depending on data distribution. Spearman’s rank correlation coefficient was utilized to evaluate the relationship between serum IL-17A levels and participants’ characteristics, disease activity parameters, and radiological findings. All statistical tests were two-sided, and a *p*-value ≤ 0.05 was considered statistically significant with a 95% confidence interval (CI).

## 3. Results

A total of 120 subjects participated in the study and were classified into three equal groups: RA, OA, and control. The mean age of the participants in the control group was 42.2 ± 7.82 years, ranging from 24 to 53 years, and approximately one-quarter of participants were male. In the RA and OA groups, the age of patients was 45.2 ± 9.33 years and 46.3 ± 7.72 years, with a predominance of females in the RA group compared to OA and control groups (*p* = 0.013). However, the difference in age among groups was not statistically significant (*p* = 0.07), as shown in Table 1.

Serum levels of IL-17 were significantly higher in the OA and RA groups compared to the control group, with mean values of 151.69 ± 23.1 in the OA group, 141.37 ± 51.09 in the RA group, and 32.46 ± 20.71 in the control group (*p* < 0.001) (Table 1).

In RA patients, serum IL-17 levels showed a statistically significant positive correlation with HAQ-DI, disease activity parameters (VAS and CDAS), ultrasonographic finding (synovitis only), and disease duration. However, no statistically significant associations were found between IL-17 and other radiological or functional parameters of RA patients, as shown in Table 2 and Figure 1a,b.

In OA patients, serum IL-17 levels showed a statistically significant positive correlation with each of the following: disease activity (VAS), ultrasonographic findings (specifically synovitis), and age. However, no significant associations were observed between IL-17 levels and other radiological or functional parameters, as shown in Table 2 and Figure 1c,d.

When comparing the disease groups (RA and OA), there was a statistically significant difference in radiological and functional parameters. In particular, the presence of synovitis and erosions was significantly higher in RA patients (Figure 2 illustrates different MSUS findings in RA group), while osteophytes were significantly more common in OA patients (*p* < 0.001; see Figure 3). However, no statistically significant differences were observed between the RA and OA groups regarding VAS scores, the presence of effusion, or serum IL-17 levels (*p* = 0.2, 0.25, and 0.26, respectively), as shown in Table 3.

## 4. Discussion

IL-17 has always been known as one of the proinflammatory cytokines [28], and previous studies implicated that the IL-17/IL23 pathway and Th17 cells play an important role in inflammation-related diseases [29,30]. IL-17’s ability to induce the expression of inflammatory chemokines and proinflammatory cytokines has been linked to increases in disease activity in RA [31].

Our results are considered innovative as this may be one of the first studies designed to assess the relationship between IL-17 serum levels, MSUS findings, and clinical disease activity in both RA and hand OA patients.

In the current study, the IL-17 level in both the RA group and the OA group increased more than 4-fold compared to the control group. In other studies conducted on Egyptian RA patients, the level of IL-17 increased 5-fold in one study [32] and 8-fold in another [19] when compared to the control.

When we compared the serum level of IL-17 between RA and OA patients, our results showed no difference between both diseases. These results were consistent with those reported by van Baarsen et al. [11]. The expression levels of IL-17 and its receptors were not different between RA, OA, and PsA patients. This was further elucidated by Hussein et al., who counted the CD4+ and CD8+ T-cell subsets and measured levels of some cytokines (TNF-α, IL1-β, IL-10, and IL-17) and a soluble intercellular adhesion molecule-1 (sICAM-1) in the sera and synovial fluids of RA and OA patients [33].

In this study, there were significant correlations between serum levels of IL-17 and disease activity parameters (CDAS and VAS) in the RA group, and our results are consistent with reports of previous studies performed on Egyptian RA patients; one study found a correlation between the serum level of IL-17 and DAS-28, ESR, CRP, and TNF-α [19,34].

Regarding the correlation between serum IL-17 and disease activity parameters in OA patients, our tests did not yield significant results, except for the VAS (pain intensity and activity measure). This suggests that the patients with a higher level of IL-17 experienced greater pain in their hand joints. Nonetheless, the functional assessment done by the AUSCAN questionnaire showed no correlation with the IL-17 serum level. In part, our results are in agreement with those of Mohamed et al. [35], but another part of our findings contrast with the findings of these authors. Mohamed et al. measured VAS and Western Ontario and McMaster Universities Osteoarthritis Index (WOMAC) values in OA patients and found a statistically significant direct correlation between IL-17 and both the WOMAC and the VAS pain scale. This finding may be explained by a very comprehensive review by Miller et al., who suggested that the pain in OA patients may be attributed to the significant role of IL-17 [36].

Mathiessen and Conaghan [37] also suggested that peripheral nociceptive neurons are sensitized by inflammation, leading to increased pain sensitivity and pain perception. On the other hand, Torres et al. [38] stated that pain may not be directly linked to IL-17 but rather to other factors resulting from OA pathological changes, such as bone attrition.

IL-17 is a proinflammatory cytokine believed to play a role in the joint inflammatory process and the development of destructive arthritis [39]. Lubberts et al. stated that IL-17 overexpression leads to the aggravation of cartilage damage [40]. These previous statements have led us to believe that most cases with IL-17 overexpression in serum will show evidence of inflammation in an MSUS examination. However, to our surprise, our results showed a significant correlation between serum IL-17 and only synovitis as MSUS evidence of inflammation.

On the other hand, we found no correlation between the ultrasound findings indicating structural damage, such as erosion, or osteophytes. This was contrary to the results published by Elhewala et al., which showed a correlation between both synovial hypertrophy and erosion in RA patients and serum levels of IL-17 in the knees, wrists, and second MCPs [41].

Our study found a more obvious inflammatory effect of IL-17 in both OA and RA rather than bone remodeling. From a clinical perspective, patients exhibiting more pronounced synovitis on simple MSUS examination, along with elevated serum IL-17 levels, may be more responsive to IL-17 blockade therapy than other treatment options.

## 5. Conclusions

In the present study, we investigated the relationship between IL-17 serum levels, clinical signs of disease activity, and radiological evidence of both activity and structural changes, as assessed by MSUS, in patients with either RA or OA. Ultrasonographic bony changes (erosions and osteophytes) did not show a correlation with serum levels of IL-17 in either the RA or OA groups. Regarding functional assessment, serum IL-17 levels were correlated with HAQ in the RA group; however, no such correlation was observed with the AUSCAN in the OA group. Based on these findings, it can be inferred that patient stratification according to IL-17 serum levels may be essential for optimizing anti-IL-17 therapy. This could explain why some patients do not respond to anti-IL-17 therapy; however, it emphasizes the need for further investigations into alternative pathways and mechanisms that may contribute to treatment failure.

## Figures and Tables

**Figure 1 diagnostics-15-01335-f001:**
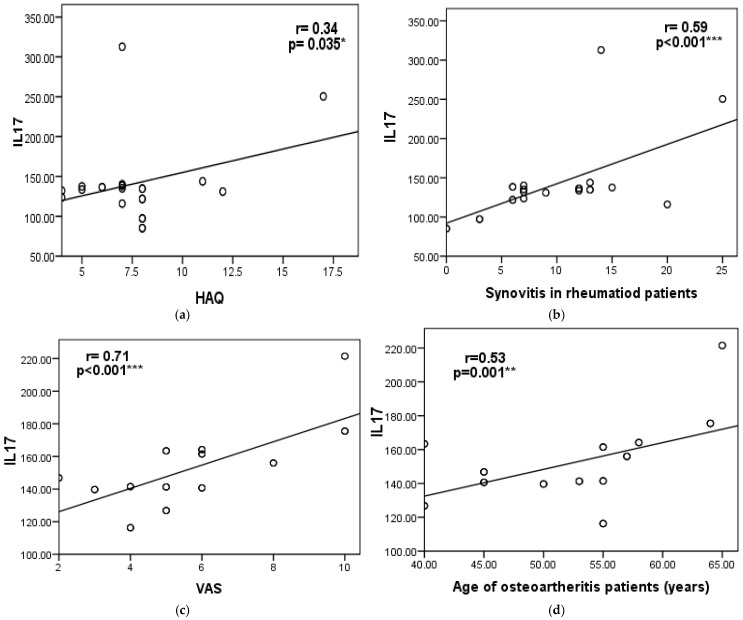
Scatter diagram showing the correlation between serum IL-17 levels and (**a**) Health Assessment Questionnaire (HAQ) and (**b**) synovitis in the RA group, and (**c**) the visual analog scale (VAS) (**d**) and age of osteoarthritis patients. Significant differences were defined as *** *p* < 0.001, ** *p* < 0.01, and * *p* < 0.05.

**Figure 2 diagnostics-15-01335-f002:**
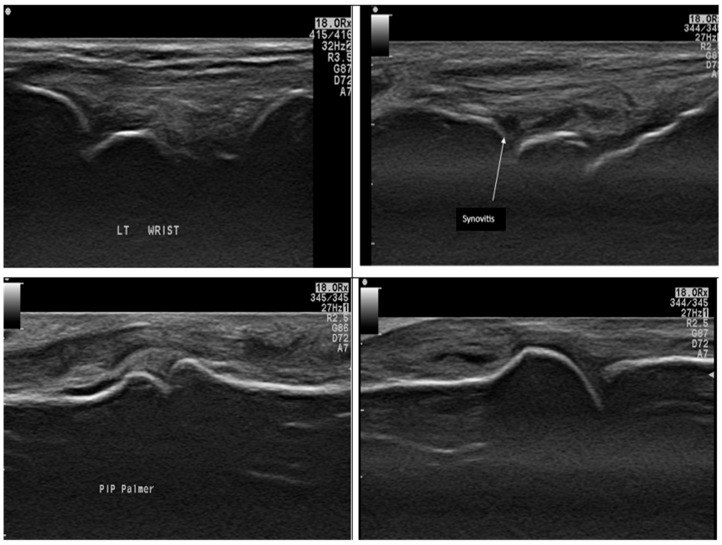
Ultrasound findings in different RA patients by a grayscale axial scan of the hand: The top two images show Grade II joint synovitis in the wrist (dorsal scan) with tenosynovitis in the 2nd case. The bottom two images show a PIP joint (Palmer scan) and an MCP joint (dorsal scan). Both show hypoechoic areas occupying joint space, indicating joint synovitis.

**Figure 3 diagnostics-15-01335-f003:**
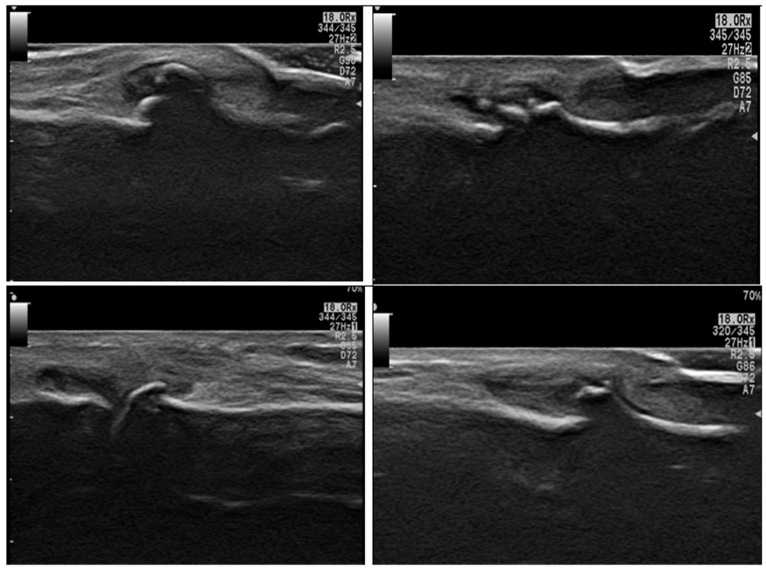
Ultrasound findings in different OA patients by grayscale axial scan of the hand: show different degrees of osteophytes and synovitis affecting DIP joints and a PIP joint (bottom right image).

**Table 1 diagnostics-15-01335-t001:** Demographic characteristics and level of IL-17 among the studied groups.

Groups		Control	RA Group	OA Group	Test of Significance	*p*-Value
	Parameter					
Age (years) -Mean ± SD -Range		42.2 ± 7.82 24–53	45.2 ± 9.33 29–61	46.3 ± 7.72 24–62	F test	0.07
Sex -Male = *n* (%) -Female = *n* (%)		10 (25.0) 30 (75.0)	2 (5.0) 38 (95.0)	12 (30.0) 28 (70.0)	χ^2^ test	0.013 *
IL-17 (pg/mL) -Mean ± SD -Range		32.46 ± 20.7 8.3–75.0	141.37 ± 51.9 85.2–312.8	151.69 ± 23.11 6.3–221.5	F test	<0.0001 *

RA: rheumatoid arthritis; OA: osteoarthritis; IL-17: interleukin-17. * Significant difference.

**Table 2 diagnostics-15-01335-t002:** Correlation of IL-17 with radiological parameters, functional parameters, and disease duration in RA and OA patients.

Variable	IL-17 in RA Group		IL-17 in OA Group	
	r	*p*-Value	r	*p*-Value
AUSCAN	NA	NA	0.124	0.46
CDAS	0.4	0.02 *	NA	NA
Erosions	0.25	0.12	−0.23	0.19
Effusion	0.06	0.72	−0.23	0.19
Osteophytes	0.006	0.97	−0.34	0.84
Disease Duration	0.37	0.018 *	0.26	0.13

RA: rheumatoid arthritis; OA: osteoarthritis; IL-17: interleukin-17. r: Pearson’s correlation coefficient; AUSCAN: Australian/Canadian Osteoarthritis Hand Index Questionnaire; NA: non-applicable; CDAS: Clinical Disease Activity Score. * Significant difference.

**Table 3 diagnostics-15-01335-t003:** The relationship between RA and OA regarding IL-17 and radiological and functional parameters.

Groups		RA Group Mean ± SD	OA Group Mean ± SD	Test of Significance	*p*-Value
	Parameter				
IL-17 (pg/mL)		141.37 ± 51.09	151.69 ± 23.13	*t* test	0.26
VAS		5.12 ± 1.79	5.7 ± 2.29	*t* test	0.21
Synovitis		9.8 ± 6.02	5.18 ± 2.37	*t* test	<0.0001 *
Erosions		1.42 ± 0.16	1.57 ± 0.37	*t* test	<0.0001 *
Effusion		1.24 ± 1.82	1.24 ± 2.61	MW	0.25
Osteophytes		6.58 ± 5.79	11.78 ± 6.68	MW	<0.0001 *

RA: rheumatoid arthritis; OA: osteoarthritis; IL-17: interleukin-17. VAS: Visual Analog Scale; MW: Mann–Whitney Test. * Significant difference.

## Data Availability

The datasets analyzed during the current study are available from the corresponding author on reasonable request.

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
