# Peer review of "Serum Interleukin-17 and Its Association with Inflammation and Bone Remodeling in Rheumatoid Arthritis and Hand Osteoarthritis: Insights from Musculoskeletal Ultrasound"

_diagnostics, 2025, doi:10.3390/diagnostics15111335_

Round 1

Reviewer 1 Report

Comments and Suggestions for Authors

The study by Ebaid et al. is an original research article examining the correlation between IL-17 and bone remodeling in Rheumatoid Arthritis (RA) and Hand Osteoarthritis (OA).

The introduction requires significant improvement to better present the current state of knowledge and clearly identify research gaps concerning the role of IL-17 in RA and OA. Without this context, the novelty and scientific value of the study are not evident.

Additionally, the manuscript lacks critical information regarding patient recruitment. The authors should specify where and when the patients were recruited and whether ethical approval was obtained form the relevant hospital. Further, who took the HAQ and other surveys? It is also important to confirm that informed consent was acquired including permission for the publication of results .

The English language throughout the manuscript should be improved. This includes correcting typographical errors and paying closer attention to punctuation—especially commas and periods—in the discussion section. As it stands, several sentences are unclear due to these issues, affecting the overall readability and interpretation of the findings.

In addition, the discussion and conclusion should more clearly communicate the implications and potential applications of the study's findings, including those on anti-IL 17 therapy. The clinical or translational relevance of the observed correlation between IL-17 and bone remodeling should be explicitly stated, as this would help underline the value of the research and guide future studies or therapeutic approaches.

Author Response

Response to Reviewer 1 Points

First of all, we appreciate the time and effort the reviewer had dedicated to providing his valuable feedback on our manuscript. 

Here is a point-by-point response to the reviewer’s points and concerns.

The study by Ebaid et al. is an original research article examining the correlation between IL-17 and bone remodeling in Rheumatoid Arthritis (RA) and Hand Osteoarthritis (OA).

Point 1: The introduction requires significant improvement to better present the current state of knowledge and clearly identify research gaps concerning the role of IL-17 in RA and OA. Without this context, the novelty and scientific value of the study are not evident.

Response: Thank you for your comment. Changes have been made according to your valuable instructions.

Point 2: Additionally, the manuscript lacks critical information regarding patient recruitment. The authors should specify where and when the patients were recruited and whether ethical approval was obtained form the relevant hospital. Further, who took the HAQ and other surveys? It is also important to confirm that informed consent was acquired including permission for the publication of results.

Response: Thank you for your observations.

  • The recruitment information has been clarified and added to the manuscript (Line 101).
  • The confirmation of informed consent has been mentioned in the Informed consent statement; however, given the use of anonymized data and absence of any personally identifiable information, consent for publication was not required (Line 314).
  • In addition, we have clarified in the manuscript that HAQs and AUSCAN were reported by patients under supervision of research staff (Line 124) while VAS and CDAS were administered and calculated by trained rheumatologists (Line 134).

Point 3: The English language throughout the manuscript should be improved. This includes correcting typographical errors and paying closer attention to punctuation—especially commas and periods—in the discussion section. As it stands, several sentences are unclear due to these issues, affecting the overall readability and interpretation of the findings.

Response: Thank you for your comment. The manuscript has been meticulously revised for English and typos errors.

Point 4: In addition, the discussion and conclusion should more clearly communicate the implications and potential applications of the study's findings, including those on anti-IL 17 therapy. The clinical or translational relevance of the observed correlation between IL-17 and bone remodeling should be explicitly stated, as this would help underline the value of the research and guide future studies or therapeutic approaches.

 Response: Thank you for your comment. The discussion and conclusion have been modified to address the implications of the study findings according to your insightful comment (Line 288 and 299).

Reviewer 2 Report

Comments and Suggestions for Authors

Comments: Manuscript addressed an important topic but manuscript may require attention on following comments.

  • Abstract: For clarity about distribution of patients, number patients under each group must be mentioned.
  • Sample size: inappropriate citations observed. Reference 22 supported only ‘IL-17 in RA patients is 204.1+33.8 pg/mL’ but reference for ‘OA patients is 180.0 ± 40.0 pg/mL’ is lacking. Proofread the manuscript to avoid inappropriate citations.  
  • CD4 Th17, CD8 Tc17 cells? Standard abbreviation should be used, e.g. CD4+ Th17, CD8+ Tc17 cells.
  • English and typos need attention, e.g. ‘pathogenesis both diseases’, ‘fourth of them were’
  • Some sentences are not supported by references. ‘can stimulate the release of TNF-α, IL-1b, and IL-6 from bone cells, synoviocytes, macrophages, and cartilage.’ Proofread the manuscript to add relevant references wherever required.
  • Table 1. Values are randomly (formatting issue) organized. Please revise appropriately. Sex male and female (values in bracket is percentage, it should be written with clarity).
  • Since gender distribution is not equal, separate comparison of IL-17 can be made based on gender.
  • Abbreviation to be expanded when first used.
  • Sensitivity (detection range) of the ELISA kit to be mentioned. No kit catalogue is mentioned to retrieve the information online.
  • Conclusion: ‘specifically in RA patients, but not in those with OA.’ Data from present study does not support the conclusion as anti-IL-17 therapy based outcome not evaluated.
  • There are previous investigations that already reported role of IL-17 in RA and OA. What is the novelty of the present study? Justification required.

Comments on the Quality of English Language

Please refer to above comments.

Author Response

 Response to Reviewer 2 Points

First of all, we appreciate the time and effort the reviewer had dedicated to providing his valuable feedback on our manuscript. 

Here is a point-by-point response to the reviewer’s points and concerns.

Comments: Manuscript addressed an important topic but manuscript may require attention on following comments.

Point 1: Abstract: For clarity about distribution of patients, number patients under each group must be mentioned.

Response: Thank you for your comment. The number under each group has been mentioned in abstract (Line 20).

Point 2: Sample size: inappropriate citations observed. Reference 22 supported only ‘IL-17 in RA patients is 204.1+33.8 pg/mL’ but reference for ‘OA patients is 180.0 ± 40.0 pg/mL’ is lacking. Proofread the manuscript to avoid inappropriate citations.  

Response: Thank you for your comment. We agree with your observation. The Reference has been added for OA patients, and the statement has been revised to cite the correct mean ± SD utilized to calculate sample size (Line 95).

Point 3:CD4 Th17, CD8 Tc17 cells? Standard abbreviation should be used, e.g. CD4+ Th17, CD8+ Tc17 cells.

Response: Thank you for your comment. We have used a standard abbreviation as suggested by reviewer (Line 42).

Point 4: English and typos need attention, e.g. ‘pathogenesis both diseases’, ‘fourth of them were’

Response: Thank you for your comment. The manuscript has been meticulously revised for English and typos errors.

Point 5: Some sentences are not supported by references. ‘can stimulate the release of TNF-α, IL-1b, and IL-6 from bone cells, synoviocytes, macrophages, and cartilage.’ Proofread the manuscript to add relevant references wherever required.

Response: Thank you for your comment. The relevant references have been added accordingly.

Point 6: Table 1. Values are randomly (formatting issue) organized. Please revise appropriately. Sex male and female (values in bracket is percentage, it should be written with clarity).

Response: Thank you for your observation. We have clarified that the value in the brackets is percentage for the specified table (Line 187).

Point 7: Since gender distribution is not equal, separate comparison of IL-17 can be made based on gender.

Response: We appreciate the suggestion regarding gender-based comparisons. However, the gender imbalance in the RA group reflects the known epidemiological pattern of RA, which predominantly affects females. Although both RA and OA show a female predominance, RA is notably more skewed, with a reported female-to-male ratio of 2–3:1. For this reason, while OA and control groups were matched for both age and sex, where disease prevalence permits such matching, the RA group was matched for age only, reflecting the true gender distribution of RA without compromising the validity or feasibility of recruitment

Furthermore, existing literature does not consistently demonstrate significant differences in IL-17 serum levels based on gender alone [1,2]. Therefore, a stratified analysis by gender in our sample is unlikely to provide meaningful or additional insights beyond the observed disease-specific differences.

Point 8: Abbreviation to be expanded when first used.

Response: Thank you for your observation. All abbreviations have been revised and expanded when first used.

Point 9: Sensitivity (detection range) of the ELISA kit to be mentioned. No kit catalogue is mentioned to retrieve the information online.

Response: Thank you for your comment. The sensitivity of ELISA kit (Detection range) was added to manuscript and kit catalogue was mentioned as well (Line 157).

Point 10: Conclusion: ‘specifically in RA patients, but not in those with OA.’ Data from present study does not support the conclusion as anti-IL-17 therapy based outcome not evaluated.

Response: Thank you for your observation, we agree with you. We have modified and rephrased the sentence to make our point of view clearer and more understood (Line 299).

Point 11: There are previous investigations that already reported role of IL-17 in RA and OA. What is the novelty of the present study? Justification required.

Response: We agree with you in your opinion that the role of IL17 in both diseases has been extensively studied. We have highlighted the gap, particularly the lack of studies focusing on the correlation between IL-17 serum levels and MSUS findings (Line 78) and direct comparisons incorporating MSUS findings (Line 83). Moreover, the extension of clinical implication of high serum IL-17 to radiological findings by MSUS has been illustrated in discussion (Line 288) and conclusion (Line 299).

  1. Meher, J.; Patel, S.; Nanda, R.; Siddiqui, M.S. Association of Serum IL-17 and IL-23 Cytokines With Disease Activity and Various Parameters of Rheumatoid Arthritis in Indian Patients. Cureus 2023, 15, e49654, doi:10.7759/cureus.49654.
  2. Farag, M.A.; El Debaky, F.E.; Abd El-Rahman, S.M.; Abd el-khalek, S.M.; Fawzy, R.M. Serum and synovial fluid interleukin-17 concentrations in rheumatoid arthritis patients: Relation to disease activity, radiographic severity and power Doppler ultrasound. The Egyptian Rheumatologist 2020, 42, 171-175, doi:https://doi.org/10.1016/j.ejr.2020.02.009.

Round 2

Reviewer 1 Report

Comments and Suggestions for Authors

Thr manuscript is markedly improved, and only update of references is suggested.

Author Response

Response to Reviewer 1 Points
First of all, we appreciate the time and effort the reviewer had dedicated to providing his valuable feedback on our manuscript.
Here is a point-by-point response to the reviewer’s points and concerns.
Point 1: The manuscript is markedly improved and only update of references is suggested.
Response: Thank you for your suggestion. the reference numbers (2, 4-8, 10, 12, 18, 20, and 29) have been added to the current version of manuscript to update the reference list.

Reviewer 2 Report

Comments and Suggestions for Authors

Comments:

  • Table 1. formatting issue still detected. Make sure to avoid such mistakes.
  • Some references were added/deleted, but revised manuscript did not keep track of that or highlighted the changes. It is difficult for reviewer to review any changes.
  • Line#168, ‘3 mL’ can be revised as ‘Three millilitre’. Since revised manuscript presented in track change mode, authors must proofread to avoid typos and sentence fragmentation issue.
  • Detection range of ELISA kit is 2.8 to 200 pg/mL, but highest values reported in RA and OA are 312.8 and 221.5. How authors detected values beyond the detection limit of the kit? Justification required. How many subjects under each group were reported to have higher values i.e. above 200pg/ml? Information must be incorporated in methodology.

Author Response

Response to Reviewer 2 Points
First of all, we apologize to the reviewer if the track-changes version of our manuscript was confusing and did not clearly demonstrate the required modifications. We appreciate the time and effort the reviewer had dedicated to providing his valuable feedback on our manuscript.
Here is a point-by-point response to the reviewer’s points and concerns.
Point 1: Table 1. formatting issue still detected. Make sure to avoid such mistakes.
Response: Thank you for your comment. The table formatting has been adjusted (Line 189).
Point 2: Some references were added/deleted, but the revised manuscript did not keep track of that or highlighted the changes. It is difficult for reviewer to review any changes.
Response: Thank you for your comment. The reference numbers (3-9, 11-16) were deleted form the initial manuscript while the reference numbers (2, 4-8, 10, 12, 18, 20, and 29) have been added to the current version of manuscript.
Point 3 Line#168, ‘3 mL’ can be revised as ‘Three millilitre’. Since revised manuscript is presented in track change mode, authors must proofread to avoid typos and sentence fragmentation issue.
Response: Thank you for your observation. We have corrected it to “Three milliliters” (Line 152) and proofread the manuscript to avoid the fragmentation issue.
Point 4: Detection range of ELISA kit is 2.8 to 200 pg/mL, but highest values reported in RA and OA are 312.8 and 221.5. How authors detected values beyond the detection limit of the kit? Justification required. How many subjects under each group were reported to have higher values i.e. above 200pg/ml? Information must be incorporated in methodology.
Response: We agree with your comment. Therefore, when we added the detection range of the assay kit, we have demonstrated that samples with concentrations exceeding the assay’ upper limit were diluted appropriately and reanalyzed to obtain accurate measurements (Line 160 to 162). However, we respectfully believe that specifying the number of subjects with values above 200 pg/mL is not directly relevant to the methodological framework, as it reflects an outcome rather than a predefined criterion or analytical step. Additionally, we did not base our analyses or conclusions on a cutoff of 200 pg/mL, nor was this threshold predefined as clinically or biologically significant within the context of our study objectives. Therefore, we have not included this information in the Methodology or Results, as we believe it does not add value to the interpretation or validity of the findings.
